# Engineered probiotic *Lactobacillus plantarum* WCSF I for monitoring and treatment of *Staphylococcus aureus* infection

Haoran Li,[1,2] Minjun Jia,[1] Qingsheng Qi,[2] Qian Wang[1]

**ABSTRACT**   *Lactobacillus plantarum* is one of the most thoroughly researched species of the genus *Lactobacillus*, which possesses the characteristics of easy genetic transformation, high-density growth, and high intestinal tract survival. *L. plantarum* has been proven to play a potential role as a probiotic delivery vector. *Staphylococcus aureus* is a common Gram-positive pathogenic bacterium. It uses an autoinducer peptide (AIP) produced by its Agr quorum-sensing (AgrQS) system to sense the population density. Using the quorum-sensing mechanism exclusive to *S. aureus*, we constructed an AgrQS system in *L. plantarum* WCSF induction and killing modules based on AIP sensing and regulation so that *L. plantarum* could effectively eliminate *S. aureus* when detecting exogenous AIP at nanomolar concentrations. By optimizing the expression strength of the two-component system AgrAC using different *L. plantarum*-derived promoters and replacing the core promoter of the AgrA-activating promoter, the activation strength of AgrQS increased from the initial 1.2-fold to 5.3-fold. By introducing the signal peptide N20-guided lysostaphin aureus protein, engineered *L. plantarum* was able to effectively control the release of lysostaphin aureus protein and inhibit the growth of *S. aureus*. For the first time, engineered *L. plantarum* can detect and treat *S. aureus* infection, laying the groundwork for the future development of engineered probiotics for the monitoring and therapy of intestinal pathogens.

**IMPORTANCE**   Bacterial infection and the emergence of drug-resistant strains are major problems in clinical treatment. *Staphylococcus aureus*, which typically infects the skin and blood of animals, is also a potential intestinal pathogen that needs to be addressed by the emergence of a new treatment approach. Probiotic therapy is the most likely alternative to antibiotic therapy to solve the problem of bacterial drug resistance in clinical practice. In this study, the engineered *Lactobacillus plantarum* can not only sense the signal AIP to detect *S. aureus* but also kill *S. aureus* by secreting the lysostaphin enzyme. Our strategy employed an Agr quorum-sensing genetic circuit to simultaneously detect and treat pathogenic bacteria, which provided a theoretical possibility for solving practical clinical bacterial infection cases in the future.

**KEYWORDS**   *Staphylococcus aureus*, self-inducing peptide, Agr quorum-sensing system, *Lactobacillus plantarum* WCSF I, lysostaphin enzyme, biosensors

In the current clinical context, bacterial infection has become an increasingly serious problem in hospitals and is one of the leading causes of the current increase in morbidity and mortality (1). Although antibiotics can effectively treat infections caused by pathogenic bacteria, they also eliminate good bacteria in the body and promote the growth of drug-resistant strains, exacerbating the effects of bacterial infection (2). Conventional procedures for diagnosing such infections and contaminations involve tedious and time-consuming sample handling, multiplex polymerase chain reactions,

Address correspondence to Qian Wang, qiqi20011983@gmail.com.

The authors declare no conflict of interest.

See the funding table on p. 13.

and biochemical assays. Thus, rapid detection and treatment methods for infection are urgently required to save lives and stop the spread of infection.

Recently, synthetic biological techniques have engineered microbes for medical diagnostics and therapeutic applications. The reprogrammed microbes, with the introduction of a sense-and-response mechanism for exogenous signals, can fight pathogens (3), treat diseases (4), deliver therapeutics (5), and tackle cancers (6). Combining the detection of quorum-sensing molecules with new metabolites produced by bacteria, modified probiotics could detect Gram-negative and Gram-positive pathogens. For instance, Huang et al. constructed a sensing and response circuit comprising the anti-*Pseudomonas aeruginosa* toxin Pyocin S5 in *Escherichia coli* Nissle 1917. In response to the acyl-homoserine lactone produced by *P. aeruginosa*, engineered bacteria released the antibiofilm enzyme dispersin B to kill pathogenic bacteria (7). Mao et al. reported a method for diagnosing *Vibrio cholerae* by modifying *Lactococcus lactis* to detect CAI-1 in pathogens (8). Moreover, *Lactobacillus reuteri* acquired the detection ability of *Staphylococcus aureus* by integrating the quorum-sensing system [Agr quorum-sensing (AgrQS)] of *S. aureus* into the modified *L. reuteri*, enabling it to sense self-inducible peptide molecule AIP-I from the nanomolar to millimolar range (9). Moreover, probiotics have been engineered previously as oral vaccine vectors for numerous bacterial and viral diseases (8).

*Lactobacillus* is one of the most prominent probiotics that can regulate the host immune response and control pathogens (10). Moreover, it is used in the fermentation of dairy products, meat, and vegetables (11, 12). *Lactobacillus plantarum* is among the most extensively studied *Lactobacillus* species (10). *L. plantarum* demonstrated an exceptionally high survival rate in the human gastrointestinal tract, and the relative survival rate of *L. plantarum* WCFS I under simulated gastrointestinal passage conditions is higher than that of other *L. plantarum* strains (13). Sasikumar et al. constructed a recombinant *L. plantarum* that can introduce oxalate decarboxylase into the body (14) to treat hyperoxaluria and calcium oxalate stone deposition, indicating that *L. plantarum* has specific relevance as a carrier of probiotics. In addition, previous studies (15) have demonstrated that *L. plantarum* WCFS I functions relatively independently and steadily in the gastrointestinal tract, regardless of the presence or absence of other bacteria (16). Correspondingly, *L. plantarum* WCFS I could survive in the gastric fluid environment with a pH of approximately 2.5, even if its concentration decreased (17). Based on the properties mentioned above, *L. plantarum* WCFS I, thus, comprises a strong candidate for the generation of new active therapeutic agents.

*S. aureus*, a Gram-positive bacteria, is one of the leading causes of clinical and nosocomial infections in humans (1, 18). Although the main infection portal of entry of the *S. aureus* is skin (19) and mucous membrane (20), *S. aureus* is the main pathogen causing cardiovascular infection and bacteremia (21) and is also a foodborne pathogen that can produce enterotoxin and cause food poisoning in humans (22). Therefore, *S. aureus* is a potential intestinal pathogen. The negative impact caused by multi-drug-resistant *S. aureus*, particularly methicillin-resistant *S. aureus*, has even surpassed the negative impact of the human immunodeficiency virus (23). New antimicrobial techniques should be developed to minimize the emergence of such resistant strains.

Self-inducible peptide (AIP) produced by the AgrQS system (24) is a clearly recognized signaling substance of *S. aureus*. The virulence factors of *S. aureus* are mainly controlled by AgrQS (25), and when the Agr was knocked out, the pathogenicity of *S. aureus* was significantly weakened (26). The AgrQS system consists of RNAII transcription driven by the $P_2$ promoter and RNAIII transcription mediated by the $P_3$ promoter (27). Four genes are encoded by the RNAII region: AgrB, AgrD, AgrC, and AgrA. Of these, AgrD encodes the AIP precursor (24), while AgrB encodes a transmembrane endopeptidase that carries out the posttranslational modification and export of AgrD outside the cytoplasmic membrane (24, 28–30). In addition, extracellular AIP is recognized by the histidine kinase AgrC, which auto-phosphorylates when AIP binds to it (31) and then phosphorylates the response regulator Agra, enabling it to bind to both $P_2$ and $P_3$ promoters, thereby

upregulating the transcription of AIP production and activating multiple downstream virulence factors (32). There are four types of AgrQS (I–IV) in *S. aureus*, and its classification is based on the primary amino acid sequence of AIP (32). AIP-I and AIP-IV differ by only one amino acid; however, the amino acid sequence and structure of AIP-II and AIP-III are dissimilar. The four AIPs can only be combined with their corresponding homologous AgrC (Fig. 1b) (33).

In this study, for the first time, we engineered *L. plantarum* WCSF I by designing the AgrQS-controlled fluorescent protein and lysostaphin expression system in *L. plantarum* to detect and inhibit the growth of *S. aureus*. The engineered *L. plantarum* WCSF I detected AIP-I in the medium and responded to the *S. aureus* supernatant by releasing the killing protein lysostaphin into the medium, effectively killing *S. aureus in vitro*. Although our proof-of-concept genetic circuit in *L. plantarum* for disease treatment is in its infancy as a therapeutic option, we believe that this synthetic biology-based antibacterial strategy using the probiotic *L. plantarum* has exciting potential to prevent pathogen invasion and treat diseases shortly.

## RESULTS

### Qualitative and quantitative analyses of AIP in *S. aureus* supernatant

To determine the type and concentration of AIP secreted by *S. aureus*, the supernatant of *S. aureus* culture after 12-h culture is analyzed by high performance liquid chromatography (HPLC) and mass spectrometry (Fig. S3). It was confirmed that AIP secreted by *S. aureus* is type AIP-I. Then, the concentration of AIP-I accumulated by *S. aureus* is calculated according to the standard curve drawn with AIP-I standard substance (Fig. S4).

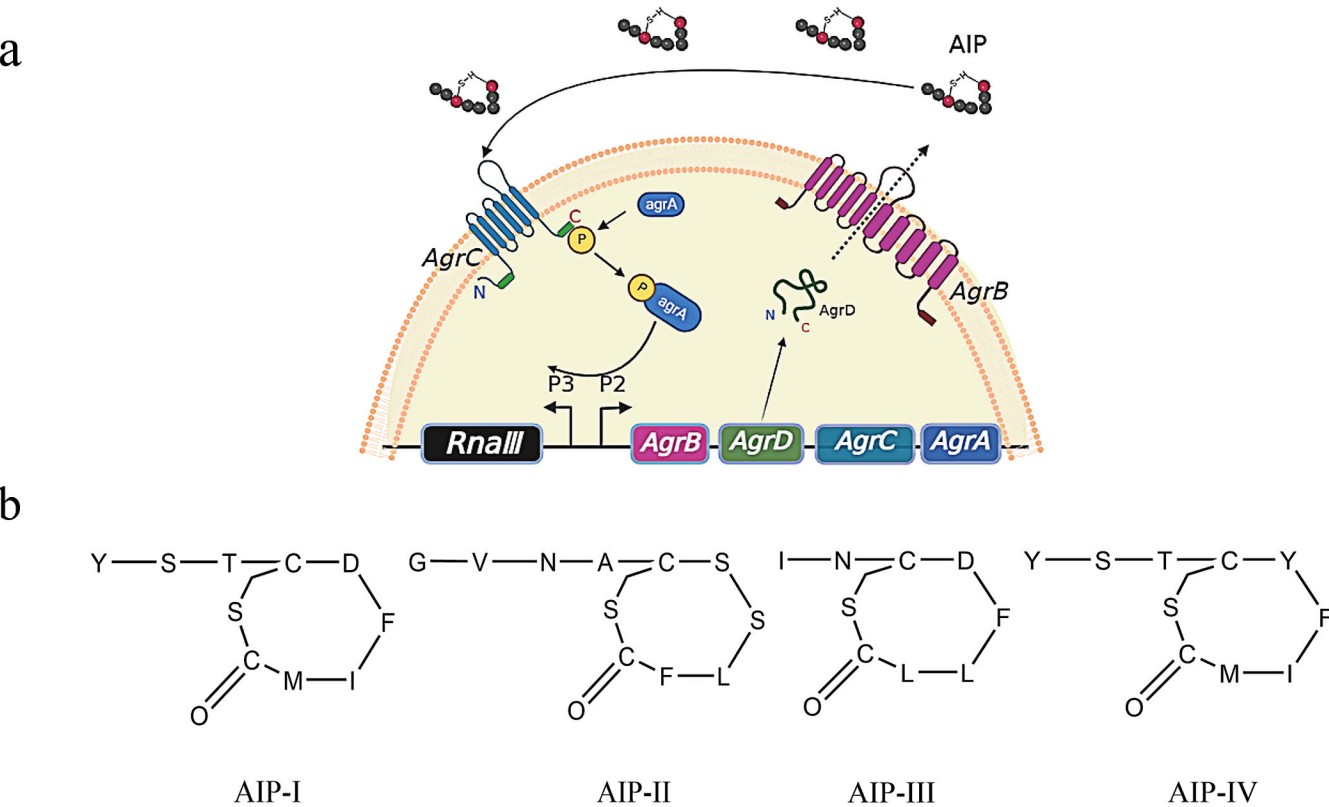

**FIG 1** Types of Agr quorum-sensing systems and self-inducible peptide structures in *Staphylococcus aureus*. (a) *S. aureus* AgrQS loop. The promoter $P_2$ regulates the Agr operon, and the promoter $P_3$ regulates the RNA III effector associated with virulence factor activation. (b) Chemical structure of self-induced proteins, AIPs I–IV.

## Construction and sensitivity detection of AIP-I responsive gene circuit in *L. plantarum* WCSF I

To detect AIP-I secreted by *S. aureus*, we designed and constructed a quorum-sensing circuit in *L. plantarum* called the AgrQS circuit with a low dosage response to AIP-I (Fig. 2a). This allowed us to detect *S. aureus*. In this designed AgrQS circuit, we selected four promoters from *Lactobacillus* species to screen the strongest promoter for the control of AgrC and AgrA (Fig. 2b): the $P_{ldhL}$ and $P_{11}$ promoters were from *L. plantarum*, the $P_{32}$ promoter was from *L. lactis*, and the $P_{slp}$ promoter was from *L. reuteri*. Simultaneously, a regulatory module containing the promoter $P_3$-regulated green fluorescent protein (GFP) gene was positioned downstream of the sensing module. Thus, AgrA that receives the phosphorylation signal can activate the GFP expression. The AgrQS circuits regulated by four different promoters were named $P_{ldhL}$-AgrQS, $P_{11}$-AgrQS, $P_{32}$-AgrQS, and $P_{slp}$-AgrQS.

To determine the efficacy of this design, engineered *L. plantarum* was cultured with 100 nM synthetic AIP-I. A microplate reader monitored the GFP fluorescence level of engineered *L. plantarum* every 10 h or so within 60 h (Fig. 2c through f). According to the characterization results, the four constructed AgrQS responded differently to the synthetic AIP-I. The induction fold of four separate promoters in the AgrQS circuits to AIP was 1.6-fold, 1.7-fold, 2.1-fold, and 1.8-fold, respectively. The $P_{32}$ promoter considerably increased the maximal fluorescence intensity of AgrQS compared to the other three promoters. However, the induction fold of the four promoters is still low and requires a substantial increase in their performance.

## Optimization of a quorum-sensing circuit in response to AIP-I

To improve the response effect of the AgrQS system toward AIP-I, we focused on engineering the AgrA-regulating promoter $P_3$. Fig. 2 depicts the high fluorescence levels of promoters $P_{slp}$ and $P_{32}$ compared to the conventional *L. plantarum* promoter $P_{ldhL}$ (Fig. 2e and f). We considered that the $P_3$ promoter was too weak, which affects the

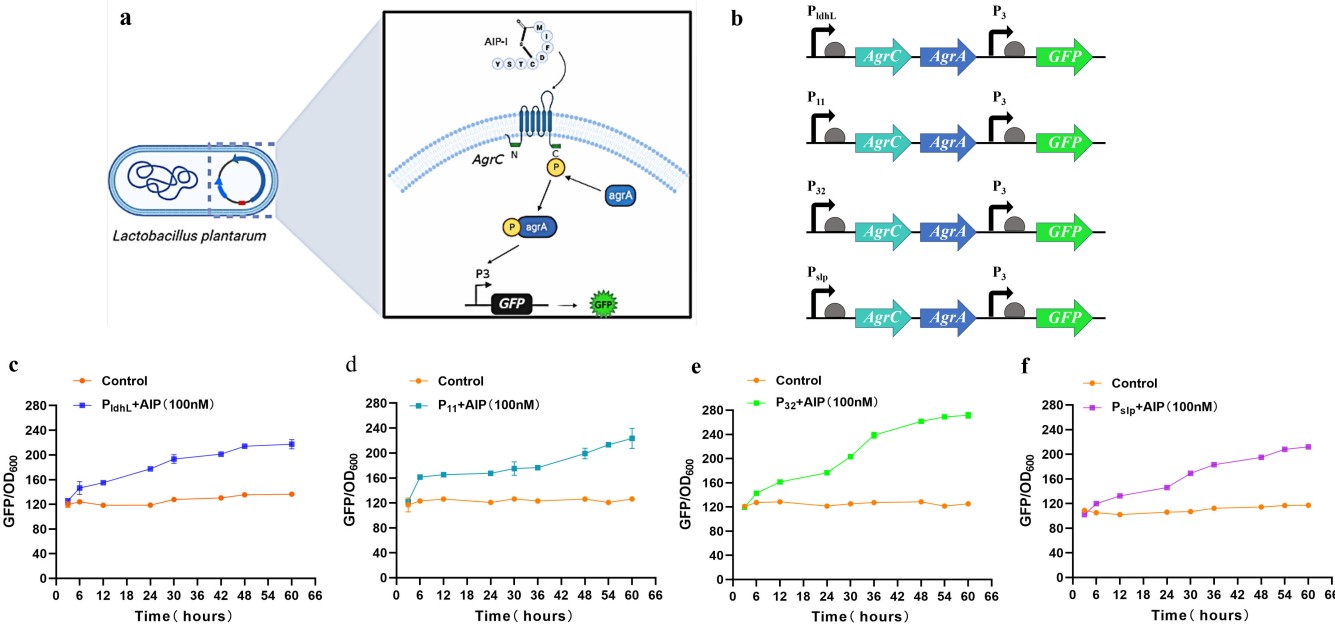

**FIG 2** Construction and characterization of the AgrQS system in *L. plantarum*. (a) The quorum-sensing elements for AIP-I detection were constructed in *L. plantarum*, including the AgrCA system for induction recognition and the $P_3$ promoter for green fluorescent protein (GFP) expression control. (b) Construction of the AgrQS induction element. The $P_{ldhL}$ and $P_{11}$ promoters from *L. plantarum*, the $P_{32}$ promoter from *Lactobacillus lactis*, and the $P_{slp}$ promoter from *Lactobacillus reuteri* regulated the AgrCA system. The $P_3$ promoter derived from *Staphylococcus aureus* regulates the GFP. (c–f) The expression of GFP is regulated by the $P_{ldhL}$, $P_{11}$, $P_{32}$, and $P_{slp}$ promoters under 100 nM AIP induction.

expression of downstream GFP. Consequently, we replaced the −35 and −10 regions of the $P_3$ promoter (34) with those of the strong promoters $P_{23}$ and $P_{11}$ while retaining the AgrA binding site (Fig. 3a). The sequence is shown in Table S1.

The modified synthetic $P_3$ promoters were labeled $P_{slp}$-$P_{11}$Mut-AgrQS, $P_{slp}$-$P_{23}$Mut-AgrQS, and $P_{32}$-$P_{11}$Mut-AgrQS. Construction of the $P_{32}$-$P_{23}$Mut-AgrQS circuit was unsuccessful. By adding 100 nM AIP-I, we confirmed the effect of the three $P_3$ promoter mutations on the activation of fluorescence expression in the AgrQS system. The results showed that the three mutant $P_3$ promoters $P_{slp}$-$P_{11}$Mut, $P_{slp}$-$P_{23}$Mut, and $P_{32}$-$P_{11}$Mut exhibited enhanced fluorescence intensity (Fig. 3b through d). The $P_{32}$-$P_{11}$Mut mutant promoter exhibited the highest fluorescence intensity, 400 a.u. After 60 h, the fluorescence intensity of $P_{32}$-$P_{11}$Mut increased approximately 3.6-fold compared to when AIP-I was not induced. The $P_{slp}$-$P_{11}$Mut mutant promoter exhibited the lowest fluorescence intensity of 300 a.u. when 100 nM AIP-I was added, and its maximum fluorescence intensity was approximately 2.5-fold than that of the basal output. In contrast, the $P_{32}$-$P_{23}$Mut mutant promoter exhibited the highest fluorescence intensity of 330 a.u., and its fluorescence intensity was increased by 3.2-fold. The results indicate that synthetic $P_3$ improved the activation strength and overall responsiveness of the AgrQS system in response to AIP-I.

## Functional characterization of AgrQS gene circuits

We evaluated the dose–response ability of the AgrQS gene circuit to AIP-I. We co-cultured different concentrations of AIP-I with engineered *L. plantarum* for 30 h. The results are shown in Fig. 4a. The GFP fluorescence intensity of the three optimized AgrQS

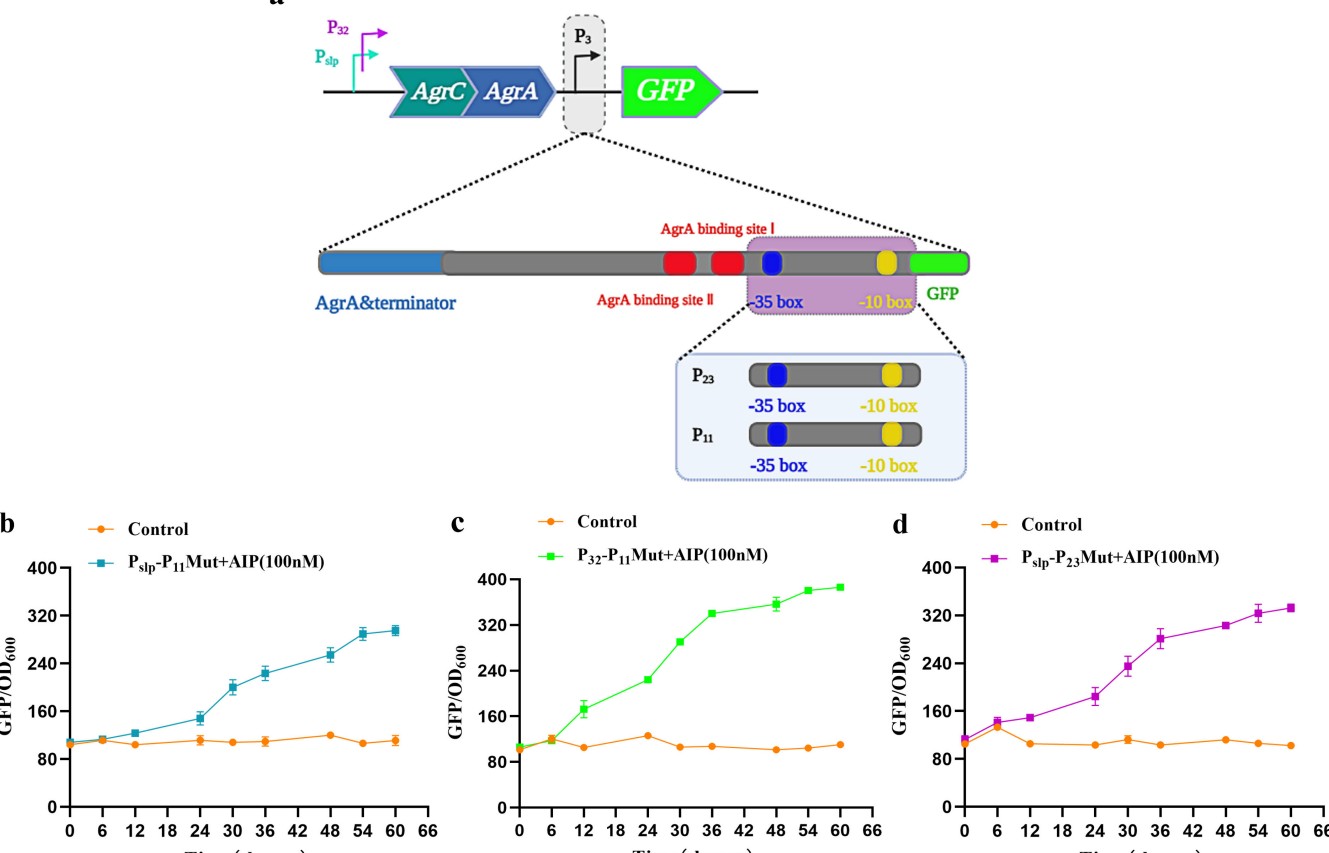

**FIG 3** Optimization and characterization of the AgrQS system in *L. plantarum*. (a) Schematic diagram of the $P_3$ promoter modification. The red box represents the sequence of the AgrA binding site, the blue box represents the −35 region, and the yellow box represents the −10 region of $P_3$. (b–d) Characterization of the AgrQS system regulated by $P_{slp}$-$P_{11}$Mut, $P_{32}$-$P_{11}$Mut, and $P_{slp}$-$P_{23}$Mut promoter combination after $P_3$ promoter mutation in *L. plantarum*.

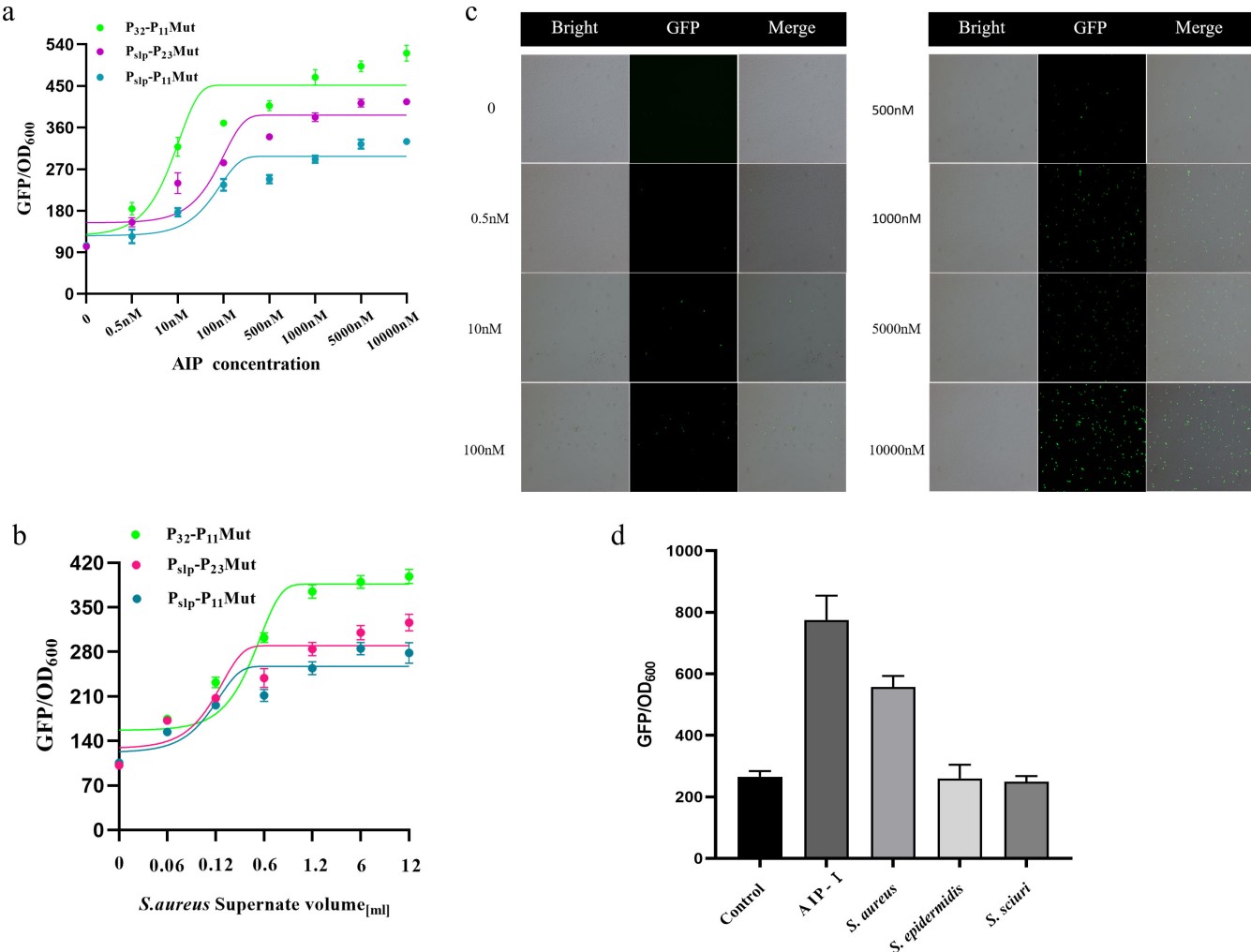

**FIG 4** Dose–response curve of constructed AgrQS system toward added AIP-I and *S. aureus* supernatant and fluorescence observation. (a) Dose–response curves of the $P_{slp}$-$P_{11}$Mut-AgrQS, $P_{32}$-$P_{11}$Mut-AgrQS, and $P_{slp}$-$P_{23}$Mut-AgrQS circuits under the induction of AIP-I with different concentrations. (b) Quantitative response curves of the $P_{slp}$-$P_{11}$Mut-AgrQS, $P_{32}$-$P_{11}$Mut-AgrQS, and $P_{slp}$-$P_{23}$Mut-AgrQS circuits under different volumes of *S. aureus* supernatant induction. (c) Fluorescence luminescence observation of engineered *L. plantarum* induced by AIP-I of 0 nM to 10,000 nM concentrations.

gene circuits showed a positive correlation trend, and the $P_{32}$-$P_{11}$Mut-GFP gene circuit showed the best response to AIP-I. When 10,000 nM AIP-I induced the $P_{32}$-$P_{11}$Mut-GFP gene circuit, the maximum fluorescence intensity was 520 a.u., and the dynamic range of fluorescence intensity was 4-fold. $P_{slp}$-$P_{11}$Mut-GFP gene circuit has the worst ability to respond to AIP-I. When AIP-I concentration is increased to 10000 nM, its maximum fluorescence intensity is 320 a.u., and the dynamic range of fluorescence intensity is 2.5-fold. The maximum fluorescence intensity of the $P_{slp}$-$P_{23}$Mut-GFP gene circuit is 370 a.u., and the dynamic range of fluorescence intensity is 3.5-fold.

Meanwhile, to assess the ability of the AgrQS gene circuit to respond to AIP-I secreted by *S. aureus*, the different volumes of *S. aureus* ATCC 6538

To verify the specificity of the constructed gene circuitry in response to signaling peptide AIP-I, we selected *Staphylococcus epidermidis* and *Staphylococcus sciuri* for comparison (Fig. 4d). The results showed that the *L. plantarum* Sensing was highly specific to AIP-I that was secreted by *S. aureus*. In addition, we evaluated the functional response of the three mutant promoters under different volumes of the supernatant solution of *S. aureus* to determine the dose–response of *S. aureus* in a realistic environment. The AgrQS circuits began to respond to *S. aureus* at 1% of the *S. aureus* supernatant and peaked at 10% of the *S. aureus* supernatant, as shown in Fig. 4b. Thus, we

have successfully constructed an AgrQS-driven monitoring system that can respond effectively to *S. aureus*.

Based on the above results, we constructed a gene circuit that can respond to the signal peptide AIP-I secreted by *S. aureus*, selected different promoters and optimized the $P_3$ promoter to control it, and screened out a gene circuit $P_{32}$-$P_{11}$Mut-GFP with the best response effect. *L. plantarum* was successfully engineered to detect the occurrence of *S. aureus* in response to the presence of AIP-I.

## Detection of the activity of engineered *L. plantarum* against *S. aureus in vitro*

The ultimate goal of developing an AgrQS gene circuit that can detect *S. aureus* is to achieve effective killing of *S. aureus*. In previous results, we successfully monitored *S. aureus* signaling molecule AIP-I through the expression of green fluorescent protein. Then, based on sensing AgrQS, we replaced the *GFP* gene in the gene circuit with the gene encoding lysostaphin endopeptidase (35) (Fig. 5a), resulting in the new AgrQS gene circuit named $P_{32}$-$P_{11}$Mut-Lys. The signal peptide N20 composed of 20 amino acids is an *N*-terminus of a cellulase (Cel-CD) and can be employed as a carrier for extracellular production of recombinant proteins (36). To improve the secretion of lysostaphin enzyme, N20 was added to the *N*-terminal of the lysostaphin endopeptidase gene to promote the secretion of lysostaphin enzyme (Fig. 5b). By adding different concentrations of AIP-I, *L. plantarum* Kill was confirmed to secrete the lysostaphin enzyme by SDS-PAGE (Fig. S7). The molecular weight of the standard lysostaphin enzyme was 27 kDa. When the concentration of AIP-I was 50 nM, the secretion of the lysostaphin enzyme began, and at 10 µM, the secretion of the lysostaphin enzyme was the highest. Meanwhile, we also quantitatively analyzed the secreted lysostaphin enzyme to determine the specific concentration of lysostaphin enzyme secreted under different induction conditions. As the concentration of synthetic AIP-I was increased from 50 nM to 10 µM, the amount of lysostaphin enzyme produced from 0.08 mg/mL to 1.2 mg/mL (Fig. S7). To further validate the bactericidal efficacy of this system, the supernatant of engineered *L. plantarum* was co-cultured with *S. aureus*, while the dynamic growth curve of *S. aureus* was continuously monitored (Fig. S9). Within 5 min, most *S. aureus* growth was inhibited at different concentrations of AIP-I (Fig. 5c). When the concentration of AIP-I reached 50 nM, the growth of *S. aureus* began to stop. At a concentration of 10,000 nM AIP-I, the $OD_{600}$ of *S. aureus* decreased from 0.88 to 0.28, which proved that the bactericidal effect of engineered *Lactobacillus plantarum* was the best (the relationship between the cell density and the absorbance at 600 nm of *S. aureus* is shown in Fig. S8). This demonstrated that our constructed AgrQS killing system had a highly effective antibacterial effect on *S. aureus*.

In addition, we calculated the size of inhibition bands on Luria-Bertani (LB) agar plates to determine the inhibitory effect of AIP-I on *S. aureus* (Fig. S10). The *L. plantarum-Kill* induced by different concentrations of AIP-I was added into the Oxford cup, and the bactericidal effect was estimated according to the diameter of the antibacterial zone (Fig. 5d). There was no bacteriostatic zone on the plate without adding AIP-I; after adding 100 nM synthetic AIP-I, the diameter was 1.3 cm; when the concentration of synthetic AIP-I was 10,000 nM, the diameter was 2.6 cm. When the *S. arueus* supernate was used as an inducer, the diameter of the bactericidal zone on the plate was 2.1 cm. These results confirmed that the strain *L. plantarum-Kill* was activated by AIP and secreted lysostaphin to kill *S. aureus* effectively.

Then, we measured the inhibitory effect of engineered *L. plantarum* supernatant on *S. aureus* biofilm formation through microporous plate experiments (Fig. 5e). It can be seen from the results that when the AIP-I induced concentration is 0.1 nM, 1 nM, and 10 nM, it hardly inhibits the formation of *S. aureus* biofilm. When AIP-I concentration was increased to 100 nM, 40% of the biofilm could be inhibited, and with increasing concentration, when the concentration was 1,000 nM, nearly 60% of the biofilm was inhibited. When the maximum concentration was increased to 10,000 nM, 85% of the *S. aureus* biofilm was inhibited.

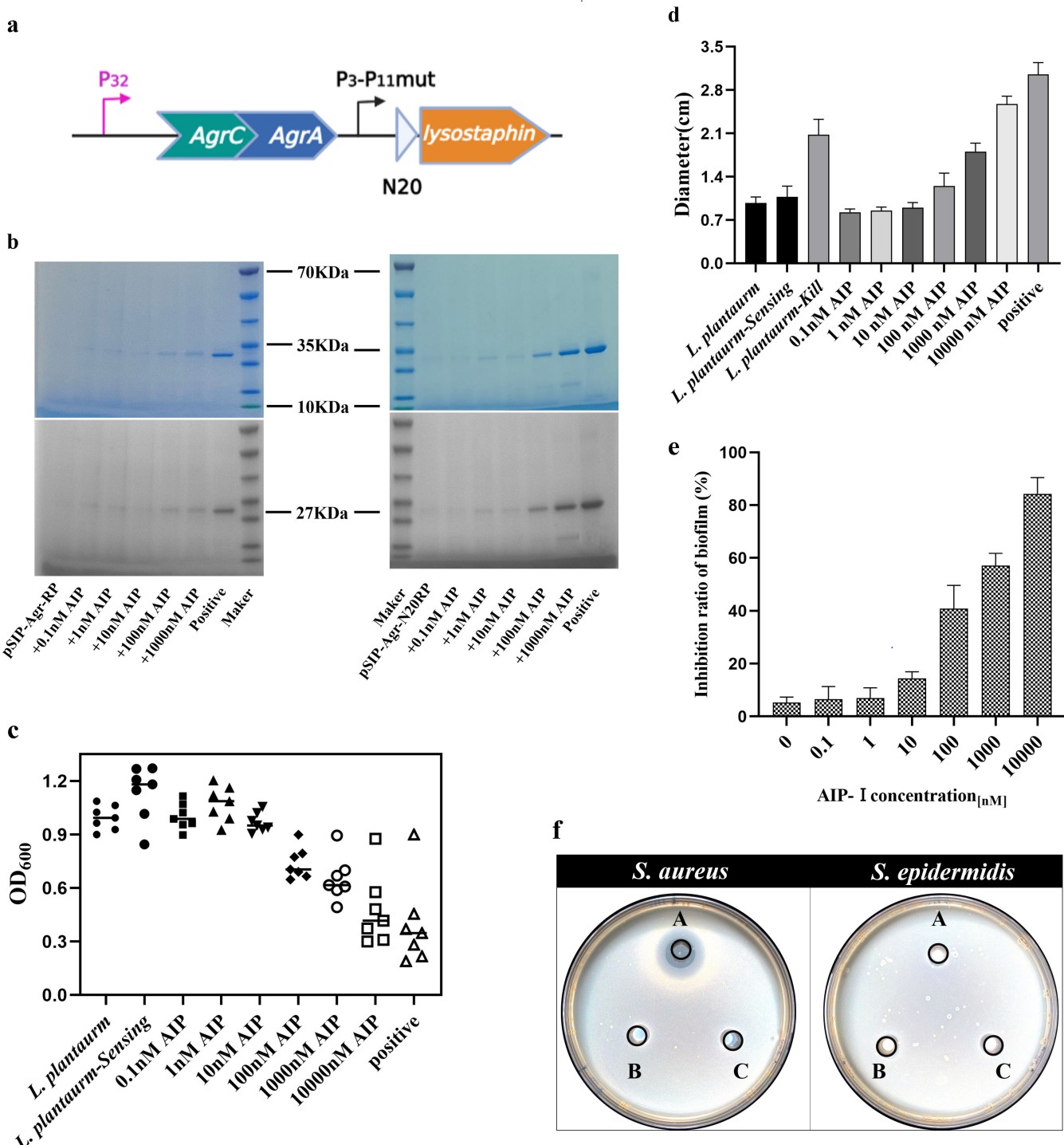

FIG 5 *S. aureus* inhibition test. (a) Modification diagram of replacing the GFP gene with the lysostaphin enzyme gene. (b) Verification of lysostaphin enzyme secretion under different induction conditions. Left, no N20 signal peptide; right, with N20 signal peptide. (c) Growth kinetics of *S. aureus*. The growth of *S. aureus* co-cultured with the *L. plantarum* supernatant. (d) Under different induction conditions, engineered *L. plantarum* inhibited *S. aureus* from forming an inhibitory zone diameter. (e) Inhibition of the ratio of the supernatant of engineered *L. plantarum* to *S. aureus* biofilm. (f) Specific bactericidal verification of killing circuit (A) *L. plantarum* Kill, (B) *L. plantarum* WCSF Ⅰ, (C) *L. plantarum* Sensing.

Finally, we verify the specificity of the constructed kill circuit. *L. plantarum* Kill, *L. plantarum* Sensing, and *L. plantarum* WCSF I were added to the Oxford cup coated with

*S. aureus* and *S. epidermidis*, respectively (Fig. 5f), and the observation results showed that only *L. plantarum* Kill could produce bactericidal rings, while others could not.

Based on these findings, we successfully engineered *L. plantarum* to monitor and inhibit *S. aureus*.

## DISCUSSION

Currently, probiotic therapy is widely investigated and applied since numerous studies have confirmed that probiotics have a significant impact on human health (37, 38). Using synthetic biology technologies, biosensors and gene circuits are developed to detect and transport therapeutic molecules, enabling the monitoring and treatment of bacterial diseases (39). For example, Kristina et al. (40) constructed a thiosulphate sensor and improved a tetracosulfate sensor in probiotic *E. coli* to detect colitis. Chen et al. developed a low cytotoxic and bacteria-targeted nanomaterial ZN-polyethoether-polye-thanolamine, which has the potential for noninvasive diagnosis of bacterial infection and can be used as an effective drug carrier for bacteria-targeted therapy (40). Although probiotic-based therapy is gaining popularity in clinical applications, the therapeutic effect is unreliable due to a lack of comprehensive understanding of the interaction between microbial hosts. Fortunately, there are already reports of engineered probiotics being used in living organisms. Ning et al. used engineered *L. lactis* on mice to detect and report the presence of *V. cholerae* in the intestinal microenvironment; *L. lactis* were able to detect *V. cholerae* through the β-lactamase reporter protein positive signal in mice feces and had no adverse effect on the survival of the mice (8). Hwang et al. introduced a quorum-sensing system that detects *N*-acylhomoserine lactone into *E. coli* Nissle 1917, enabling the modified probiotics to specifically recognize, migrate, and kill *P. aeruginosa* in mice (3). These experimental studies provided theoretical support for the safety of engineered probiotics in living intestines. In addition, the engineered probiotics lack an artificially developed method for accurate detection and treatment of bacterial infections (40).

In terms of the use of probiotics, *L. plantarum* was a good selection not only due to its certain probiotic effect and clear research background but also due to the reliable and stable delivery effect (15). Carla et al. found that the transmission frequency of *L. plantarum* R-plasmids toward the common Enterobacteriaceae was reduced under the action of metabolites, confirming that the ability to interfere with the transfer ability of R-plasmid may be a characteristic of the tested *L. plantarum* strain (41).

The probiotic *L. plantarum* WCSF I has been developed to detect and treat pathogenic bacteria for the first time in this study. We designed and constructed an AgrQS system of *S. aureus* in *L. plantarum* WCSF I to respond to the signal molecule AIP-I secreted by pathogenic *S. aureus*. To make the AgrQS system successful in *L. plantarum*, we first optimized the expression strength of the two-component system AgrAC using different *L. plantarum*-derived promoters: $P_{ldhL}$, $P_{11}$, $P_{32}$, and $P_{SPL}$. Then, the AgrA-activating promoter $P_3$ was improved by replacing its core promoter (the area between −35 and −10) with the *L. plantarum*-derived promoters $P_{11}$ and $P_{23}$. In the presence of 100 nM AIP-I, the fold change of the optimized AgrQS sensor increased from the initial 1.2-fold to 5.3-fold. After improving the performance of the AgrQS sensor *L. plantarum*, we engineered *L. plantarum* as the carrier to deliver an inhibitor of *S. aureus* and constructed an AgrQS-driven inhibitor system by replacing the lysostaphin aureus enzyme gene with the GFP gene. This system could inhibit *S. aureus* effectively in the presence of 50 nM AIP-I. The engineered *L. plantarum* was then used to inhibit the growth of *S. aureus*, providing a reference application of microbial methods for the clinical treatment of *S. aureus* infection and enhancing the clinical treatment of *S. aureus*. The biosensor could further develop chemotactic systems to target the colonization area of *S. aureus* more precisely.

In conclusion, we effectively engineered *L. plantarum* for the first time to monitor and inhibit *S. aureus*, which showed greatly amplified regulatory signals. This establishes a theoretical basis for the future prevention and treatment of this pathogen. This system

considerably improved the pathogen therapeutic efficacy of *L. plantarum*, and thus, we believe it will be helpful for both fundamental studies and future clinical applications.

## MATERIALS AND METHODS

### Bacterial strains, culture medium, and growth conditions

All bacterial strains, plasmids, and primers in this study are listed in Table 1. *S. aureus* and *E. coli* were cultured in LB agar or LB broth medium in a shaking bed of 220 rpm at 37°C. *L. plantarum* was cultured in de Man, Rogosa, and Sharp (MRS) agar medium or MRS broth medium at 37˚C. The antibiotics used were erythromycin 50 mg/mL (LB medium) and 5 mg/mL (MRS medium).

### Plasmid construction and transformation of *L. plantarum*

The *S. aureus* genome was extracted using a genome extraction kit, followed by the use of primers AgrC-F, AgrC-R, and AgrA-F. AgrA-R was used to amplify AgrC (Gen-Bank: ABX60402.2) and AgrA (GenBank: ABW06459.1) fragments from the *S. aureus* genome . The 3′ end of the AgrC-R primer and the 5′ end of the AgrA-F primer contained homologous arms which can be fused into a complete fragment. The amplified fragments of AgrC, AgrA, and GFP were incorporated into the shuttle vector pSIP403, which carried the target component promoter, $P_3$ promoter, and erythromycin resistance fragment. It was possible to clone and replicate the vector in *E. coli* XL1-Blue. All fragments were transformed into XL1-Blue, and the sequences were verified to complete the plasmid construction. The lysostaphin enzyme (GenBank: RGP42019.1) fragments were amplified using primers of Lysostaphin-F and Lysostaphin-R.

The recombinant shuttle vector in XL1-Blue was expressed in *L. plantarum*. The recombinant shuttle carrier was transformed into *L. plantarum* via electroporation using the electro-transformation technique. The electric shock procedure included 2,000 V, 25 µF, and 200 Ω, and the electric revolution constant was approximately 7.0, adding 1 mL of the MRS medium with sucrose and magnesium chloride medium, standing at 30°C for recovery culture for 1 h, coating with a solid MRS plate containing 5 mg/mL erythromycin, incubating at 37°C, and then sequencing verification.

### *S. aureus* supernatant analysis

The supernatant of *S. aureus* was tested to determine the type of AIP released by the strain used in this experiment. Overnight-cultured *S. aureus* was centrifuged for 2 min at 12,000 rpm. The supernatant was filtered using a 0.2 µM and sterile filter membrane and analyzed and identified using liquid chromatography-mass spectrometry on a C18 column (4.6 * 150 mM, 5 µm). A 33-mL sample solution was injected into the column and separated for 25 min with 20% acetonitrile and 0.1% formic acid to determine the type of AIP secreted by *S. aureus* by molecular weight (Fig. S1).

### AIP stock solutions

According to the identification results, AIP-I [YST-c(CDFIM)] was purified and synthesized by Nanjing Yuan-Peptide Biotechnology Co. (Fig. S2), and the synthetic AIP-I was dissolved in 80% $H_2O$ and 20% acetonitrile solution.

### Fluorescence detection

The colonies were incubated overnight at 37°C in the MRS medium with erythromycin. The following day, the colonies were inoculated with fresh MRS medium containing erythromycin at a rate of 2% and divided into 48-well plates. After the samples attained $OD_{600}$ values between 0.5 and 0.8, different concentrations of synthetic AIP-I were added at a rate of 2%. After removing the supernatant, the bacteria were suspended in

the phosphate buffer solution and inoculated into 48-well plates. The $OD_{600}$ and GFP fluorescence values were measured using Bio Tek Synergy enzyme-label equipment with an excitation wavelength of 485 nm and an emission wavelength of 528 nm. Using a Nikon-positive fluorescent microscope, the expression of GFP in *L. plantarum* was observed.

## Detection of lysostaphin enzyme production

*L. plantarum* containing the recombinant plasmid pSIP-P32-P11Mut-LysP (*L. plantarum* Kill) was cultured to an $OD_{600}$ value between 0.6 and 0.8, and different concentrations of synthetic AIP-I were added at a rate of 2% for overnight induction culture. The samples were centrifuged for 1 min at 12,000 rpm, and the supernatant was collected and mixed with a 5× protein electrophoresis loading buffer. The samples were heated for 5 min at 100°C. After brief centrifugation, the samples were subjected to sodium dodecyl sulfate-polyacrylamide gel electrophoresis. The samples were stained with Coomassie brilliant blue following electrophoresis.

## Lysostaphin concentration detection

According to the instructions in the bicinchoninic acid protein concentration assay kit (purchased from Beyotime Biotechnology Co., Ltd.), we quantitatively analyzed the lysostaphin enzymes produced by engineered *Lactobacillus plantarum* under different induction conditions.

**TABLE 1** Bacterial strains, plasmids, and primers were used in this study

| Name | Description | References or sources |
| --- | --- | --- |
| Strains | | |
| *S. aureus* ATCC 6538 | Secretion of type I AIP | Lab stock |
| *L. plantarum* WCSF I | Wild type | Lab stock |
| *S. epidermidis* | Wild type | Lab stock |
| *S. sciuri* | Wild type | Lab stock |
| *E. coli* XL1 Blue | Cloning strain | Life Technologies |
| *L. plantarum* Sensing | *L. plantarum* with $P_{32}$-$P_{11}$Mut-AgrQS circuit | This study |
| *L. plantarum* Kill | *L. plantarum* with $P_{32}$-$P_{11}$Mut-Lys circuit | This study |
| Plasmids | | |
| pSIP403 | *Escherichia coli* and *Lactobacillus* shuttle vector carrying erythromycin resistance | Lab stock |
| pSIP-$P_{32}$-Agr-GFP | Derivative of the pSIP403 vector with the $P_{32}$ promoter from *L. lactis* and with AgrAC genes from ATCC 6538 downstream of a gene encoding the green fluorescent protein | This study |
| pSIP-$P_{slp}$-Agr- GFP | Derivative of the pSIP403 vector with the $P_{slp}$ promoter from *L. reuteri* and with AgrAC genes from ATCC 6538 downstream of a *GFP* gene | This study |
| pSIP-$P_{32}$-$P_{11}$Mut | Derivative of pSIP-$P_{32}$-Agr-GFP with the sequence of −10 and −35 regions of the $P_3$ promoter replaced by the −10 and −35 regions of the $P_{11}$ promoter | This study |
| pSIP-$P_{32}$-$P_{11}$Mut-LysP | Derivative of pSIP-$P_{32}$-$P_{11}$Mut with the GFP gene replaced by the lysostaphin gene | This study |
| Primers (5′→3′) | | |
| AgrC-F | ATGGAATTATTAAATAGTTATAATTTTG | This study |
| AgrC-R | AGTTGAAATTATTAACAACTAG | This study |
| AgrA-F | ATGAAAATTTTCATTTGCGAAGACGATCCA | This study |
| AgrA-R | TTATATTTTTTTAACGTTTCTCACCGATGCATAGCA | This study |
| GFP-F | ATGTCAAAGGGTGAAGAATTATTTACGGGC | This study |
| GFP-R | ATTACGCATGGTATGGATGAATTGTATAAATA | This study |
| Lysostaphin-F | TTTATATATCCATGGAAGGAAACACTVGTGAAAGA | This study |
| Lysostaphin-R | TTATTTAATGGTGCCCCACAGCACGCCCAGGGTA | This study |

## Detection of growth kinetics of *S. aureus*

*L. plantarum* Kill was cultured with an $OD_{600}$ value of 0.6–0.8, and synthetic AIP-I with different concentrations was added at a rate of 2% for overnight induction culture. The samples were centrifuged for 1 min at 12,000 rpm, and the supernatant was collected. The collected supernatant was co-cultured for 45 min with *S. aureus* that had been precultured with an $OD_{600}$ value between 0.8 and 0.9, and the $OD_{600}$ value of *S. aureus* was determined using a BioTek Synergy enzyme-label instrument.

## Anti-biofilm experiment of engineering *L. plantarum* supernatant

Single colonies of *S. aureus* were selected and inoculated into test tubes containing 5-mL tryptic soy broth medium and cultured at 37°C and 220 rpm for 10 h. *S. aureus* was transferred to 96-well plates at a 1% inoculation rate, and five parallel groups were set up. The supernatant of *L. plantarum* induced by different concentrations of AIP-I was added to make total volume at 200 μL to each culture hole and was static at 30°C for 24 h. The supernatant was removed, and the bacteria were washed with sterile water three times. The 0.1% crystal violet solution was added into each hole and static culture at 37°C for 10 min. The 0.1% crystal was removed, and the bacteria were washed with sterile water three times. The 30% acetic acid solution was added into each hole to observe and determine the absorption value at 560 nm wavelength.

The inhibition ratio of biofilm was calculated as follows:

$$\text{Inhibition ratio of biofilm } (\%) = \frac{OD\ of\ Control\ Group\ -OD\ of\ Test\ Group}{OD\ of\ Control\ Group\ -OD\ of\ Blank\ Group} \times 100\%$$

## Inhibition zone method

*S. aureus* precultured with an $OD_{600}$ value of 0.8–0.9 was coated in a nonresistant LB agar medium, where an Oxford cup was placed. The *L. plantarum* Kill induced by different concentrations of AIP-I was added into the Oxford cup, and the bactericidal effect was estimated according to the diameter of the antibacterial zone.

## ACKNOWLEDGMENTS

We would like to thank Sen Wang, Xiaomin Zhao, and Yuyu Guo from the State Key Laboratory of Microbial Technology of Shandong University for their help and guidance in fluorescence microscopy. We thank the ACS authoring services team for editing the language of the draft of this manuscript.

This work was supported by grants from the National Key R&D Program of China (2019YFA0904900), the National Natural Science Foundation of China (32270089), and the Key R&D Program of Shandong Province (2020CXGC010602).

H.L.: formal analysis (lead), investigation (lead), methodology (lead), writing—original draft (lead). M.J.: formal analysis (equal), investigation (equal), methodology (equal). Q.Q.: project administration (equal), supervision (equal). Q.W.: conceptualization (lead), project administration (lead), supervision (lead), writing of original draft (lead), writing—review and editing (lead). All authors have read and approved the final manuscript.

## AUTHOR AFFILIATIONS

[1]National Glycoengineering Research Center, Shandong University, Qingdao, China
[2]State Key Laboratory of Microbial Technology, Shandong University, Qingdao, China

## AUTHOR ORCIDs

Qingsheng Qi  http://orcid.org/0000-0001-9015-4561
Qian Wang  http://orcid.org/0000-0003-3135-9385

## FUNDING

| Funder | Grant(s) | Author(s) |
|---|---|---|
| MOST | National Natural Science Foundation of China (NSFC) | 32270089 | Qian Wang |

## AUTHOR CONTRIBUTIONS

Haoran Li, Conceptualization, Data curation, Formal analysis, Investigation, Writing – original draft | Minjun Jia, Data curation, Formal analysis, Investigation | Qingsheng Qi, Supervision, Validation, Writing – original draft | Qian Wang, Funding acquisition, Supervision, Visualization, Writing – original draft, Writing – review and editing

## DATA AVAILABILITY

All data generated or analyzed during this study are included in this published article and its supplemental material.

## ADDITIONAL FILES

The following material is available online.

### Supplemental Material

**Fig. S1 to S10, Tables S1 (Spectrum01829-23-s0001.docx).** Specific instructions are indicated in the document.

### Open Peer Review

**PEER REVIEW HISTORY (review-history.pdf).** An accounting of the reviewer comments and feedback.

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
