## [Reviewer comments · Microbiology Spectrum]

Microbiology Spectrum

Engineered probiotic *Lactobacillus plantarum* WCSF I for monitoring and treatment of *Staphylococcus aureus* infection

Haoran Li, Minjun Jia, Qingsheng Qi, and Qian Wang

Corresponding Author(s): Qian Wang, Shandong University - Qingdao Campus

Review Timeline:

Submission Date:	May 3, 2023
Editorial Decision:	June 30, 2023
Revision Received:	September 2, 2023
Accepted:	September 23, 2023

Editor: Harold Marcotte

Reviewer(s): The reviewers have opted to remain anonymous.

Transaction Report:

DOI: <https://doi.org/10.1128/spectrum.01829-23>

June 30, 2023

Dr. Qian Wang
Shandong University - Qingdao Campus
Life Science School
27, shanda south road
571, north building, Life Science School
Jinan, Shandong 250100
China

Re: Spectrum01829-23 (Engineered probiotic *Lactobacillus plantarum* WCSF I for monitoring and treatment of *Staphylococcus aureus* infection)

Dear Dr. Qian Wang:

Thank you for submitting your manuscript to Microbiology Spectrum.

When submitting the revised version of your paper, please provide (1) point-by-point responses to the issues raised by the reviewers as file type "Response to Reviewers," not in your cover letter, and (2) a PDF file that indicates the changes from the original submission (by highlighting or underlining the changes) as file type "Marked Up Manuscript - For Review Only". Please use this link to submit your revised manuscript - we strongly recommend that you submit your paper within the next 60 days or reach out to me. If you wish to extend the revision period in order to address some of the reviewer's comments, please contact me.

. Detailed instructions on submitting your revised paper are below.

Link Not Available

In addition to carefully considering the reviewers' comments, ensure thorough attention is given to the method description, including (but not limited to) the description of vector construction and the rationale for promoter selection, the composition of MRSSM medium, and the bacterial inhibition test. The material and methods description should enable the study to be replicated. Additionally, please double-check the concentrations of reagents, as there appears to be an error in the reported concentrations of antibiotics in LB and MRS (50 mg/ml and 5 mg/ml) on line 248. It would also be in your best interest to improve the writing. I recommend that you ask a colleague of yours who is a native English speaker to read and provide you some feedback on the writing. You are also welcome to use one of the services here: <https://journals.asm.org/content/language-editing-services>

Sincerely,

Harold Marcotte

Journals Department
Reviewer comments:

Reviewer #1 (Comments for the Author):

Qian Wang et al. achieved significant success in inhibiting the growth of *S. aureus* by introducing a signal peptide N20 to guide the release of lysostaphin aureus protein in engineered *L. plantarum*. By utilizing various promoters and switching the core promoter for the AgrA-activating promoter, they were able to significantly increase the production of lysostaphin aureus protein. This increase in production levels was a critical factor in the successful inhibition of *S. aureus* growth.

While the manuscript's topic is important and the results are promising, the analysis and writing require significant improvements. To enhance the manuscript's impact and value, the author should consider providing more detailed explanations for their experimental design and findings. Expanding upon their methods and clearly defining the significance of their results would help readers better understand the research and its potential applications.

Major comments:

1. To further support their findings, the authors should consider conducting an HPLC analysis of both synthetic AIP and AIP secreted by *S. aureus* simultaneously. By comparing the retention time and MS/MS fragmentation of the two samples, the author can provide additional evidence that the AIP secreted by *S. aureus* is type I. Additionally, the author should provide a more detailed description of the identification procedure used in their study to ensure readers fully understand the process.
2. The authors should clearly explain the differences and relationships between the bactericidal (Fig.5c) and bacteriostatic (Fig.S7) effects observed in their study to help readers better understand the results. Additionally, the author should provide more detailed and clear information on "The bacterial inhibition test" to help readers understand the methodology used in the study.

Minor comments:

3. It would be helpful for the authors to explain the reasoning behind their selection of the PldhL, P11, P32, and Pslp promoters. This would give readers a better understanding of why these specific promoters were chosen over others.
4. The authors should provide an explanation for their selection of the -35 and -10 regions of the P3 promoter as well as promoters P23 and P11. This would give readers more insight into why these specific regions and promoters were chosen for the study.
5. The authors should clarify whether the signal peptide N20 only boosts lysostaphin enzyme expression or if it also affects the quality of the enzyme. Additionally, the author should perform further tests to confirm the exact impact of N20 on lysostaphin enzyme expression and quality.
6. The authors should ensure that the description in Table 1 is centered and that there are no alignment issues in the table. Additionally, the author should provide clearer images of Fig. 2 and increase the fluorescence in Fig.4 C to make it more visible.

Reviewer #2 (Comments for the Author):

The paper of Haoran Li et al. described the creation of an engineered probiotic strain, *Lactobacillus plantarum* WCSF I for monitoring the treatment of *S. aureus* associated infections. The authors constructed an AgrQS system in *L. plantarum* capable of inhibiting *S. aureus* when detecting exogenous AIP nanomolar concentrations. According to my opinion, although the topic is interesting it is necessary to clarify some aspects and add some experiments to improve the paper as well as the understanding by the reader.

Major comments:

1. In the abstract the authors wrote that the engineering probiotics will be useful for monitoring the intestinal pathogens. *S. aureus* is an opportunistic pathogen associated with a wide range of infections, therefore, why do not set up an engineered probiotic strain particularly versus Enterobacteriaceae (like *E. coli*, *Salmonella*) or *Listeria* etc? Please specify your choice.
2. Did the authors evaluate if the constructed could be subjected to the horizontal gene transfer? Please add some comments regarding this aspect.
3. Did the authors evaluate a possible effect against other *Staphylococcus* species belonging to the microbiota? Please, test other *Staphylococcus* species to demonstrate the selectivity.
4. Did the authors evaluate a possible anti-biofilm activity of the supernatant produced by the *L. plantarum* modified strain? Please, demonstrate the quorum sensing inhibition.
5. Regarding the bacterial inhibition test (lines 295-305): the methods is just a qualitative method. Is there a method to quantify the component associated with the inhibition (lysostaphin)?
6. In literature a similar approach has been used in *L. reuteri*. Please, add comments regarding the paper "Programming probiotic *L. reuteri* as biosensor for *S. aureus* derived AIP 1 detection" by Lubkowicz et al. (2018).

7. Did the authors have any data regarding the safety of the modified strain on intestinal cell lines?
8. A more thorough treatment of the materials and methods would be appreciated for a better understanding by the reader. The section material and methods is very confused.

Minor comments:

1. Please, specify the number of CFU/ml corresponding to the OD concentration in the section material and methods.
2. Please write in italics the bacterial genus and species in the text and in the references.
3. In the discussion the authors should explain the possible clinical applications deriving from the use of the modified strain.
4. The quality of the images must be improved. The Figures are too much small and the quality is unfortunately bad.
5. Please add the number of independent experiments performed for the sections bacterial inhibition test and *S. aureus* supernatant analysis.

Staff Comments:

Preparing Revision Guidelines

Please return the manuscript within 60 days; if you cannot complete the modification within this time period, please contact me. If you do not wish to modify the manuscript and prefer to submit it to another journal, please notify me of your decision immediately so that the manuscript may be formally withdrawn from consideration by Microbiology Spectrum.

Dear Editor,

Thank you for your letter regarding our manuscript entitled “**Engineered probiotic *Lactobacillus plantarum* WCSF I for monitoring and treatment of *Staphylococcus aureus* infection**” which was submitted for publication. We appreciate the valuable comments and concerns made by the editor and reviewers. We have carefully considered the comments and thoughtful suggestions, responded to these suggestions point-by-point, and revised the manuscript accordingly. All changes made to the text are in blue so that they may be easily identified. We hope that our response is satisfactory, and we look forward to hearing from you soon.

With regards,

Qian Wang

Comments from the editors and reviewers:

-Reviewer 1

- Qian Wang et al. achieved significant success in inhibiting the growth of *S. aureus* by introducing a signal peptide N20 to guide the release of lysostaphin aureus protein in engineered *L. plantarum*. By utilizing various promoters and switching the core promoter for the AgrA-activating promoter, they were able to significantly increase the production of lysostaphin aureus protein. This increase in production levels was a critical factor in the successful inhibition of *S. aureus* growth.

While the manuscript's topic is important and the results are promising, the analysis and writing require significant improvements. To enhance the manuscript's impact and value, the author should consider providing more detailed explanations for their experimental design and findings. Expanding upon their methods and clearly defining the significance of their results would help readers better understand the research and its potential applications.

1. To further support their findings, the authors should consider conducting an HPLC analysis of both synthetic AIP and AIP secreted by *S. aureus* simultaneously. By comparing the retention time and MS/MS fragmentation of the two samples, the author can provide additional evidence that the AIP secreted by *S. aureus* is type I. Additionally, the author should provide a more detailed description of the identification procedure used in their study to ensure readers fully understand the process.

✓ We conducted the HPLC and MS analysis of both synthetic AIP and AIP secreted by *S. aureus* respectively. MS/MS fragmentation of the two samples evidenced that the AIP secreted by *S. aureus* is a type I AIP. We have given a more detailed description of the identification procedure and revised the manuscript in blue font in the first paragraph of the first part of the results. The specific results are presented in Figure S1-S4.

2. The authors should clearly explain the differences and relationships between the bactericidal (Fig.5c) and bacteriostatic (Fig.S7) effects observed in their study to help readers better understand the results. Additionally, the author should provide more detailed and clear information on "The bacterial inhibition test" to help readers understand the methodology used in the study.

- ✓ Thank you for your advice. In this study, the effect of engineered *L. plantarum* on *S. aureus* is basically bactericidal. The lethal capacity of *L. plantarum* on *S. aureus* was verified through both measuring cell density and inhibition zone method. To help readers better understand the results, we do not use the word bacteriostatic but use the words bactericidal killing or lethal in the text. The “bacteriostatic circle test” was changed to the “inhibition zone method”. We provided more detailed and clear information on the "inhibition zone method" to help readers understand the methodology used in the study. We have moved Figure 5c of the original manuscript to Figure S9 of the current manuscript, the "inhibition zone method" in the original manuscript Figure S7 has changed into Figure S10 in the current manuscript.

Minor points:

3. It would be helpful for the authors to explain the reasoning behind their selection of the P_{ldhL} , P_{11} , P_{32} , and P_{slp} promoters. This would give readers a better understanding of why these specific promoters were chosen over others.

- ✓ The four promoters were derived from similar *Lactobacillus* species and were compared by the expression strength to obtain an optimal promoter. We have explained the reason for the selection of these promoters. We revised the manuscript in blue font in the second paragraph of the second part of the results.

4. The authors should provide an explanation for their selection of the -35 and -10 regions of the P_3 promoter as well as promoters P_{23} and P_{11} . This would give readers more insight into why these specific regions and promoters were chosen for the study.

- ✓ We have explained the selection of the -35 and -10 regions of the P_3 promoter as well as promoters P_{23} and P_{11} . We revised the manuscript in blue font in the first paragraph of the third part of the results.

5. The authors should clarify whether the signal peptide N20 only boosts lysostaphin enzyme expression or if it also affects the quality of the enzyme. Additionally, the author should perform further tests to confirm the exact impact of N20 on lysostaphin enzyme expression and quality.

- ✓ The signal peptide N20 is an N-terminus of a cellulase (Cel-CD) and can be employed as a carrier for extracellular production of recombinant proteins (Gao et al. Microbial Cell

Factories (2015) 14:49). The aim of using N20 in this paper is to improve the secretion of lysostaphin enzyme outside of the cells and enhance the killing of *S. aureus*. We have performed further tests to confirm the exact impact of N20 on the lysostaphin enzyme secretion and quality in blue font in the first paragraph of the fifth part of the results (Page7, line 6 to line 9), and compared the function of our system with or without the addition of N20 in Figure 5b.

6. The authors should ensure that the description in Table 1 is centered and that there are no alignment issues in the table. Additionally, the author should provide clearer images of Fig. 2 and increase the fluorescence in Fig. 4 C to make it more visible.

✓ We have centered the description to ensure no alignment issues in Table 1 and improved the pixel quality in Figure 2 and Figure 4c.

-Reviewer 2

The paper of Haoran Li et al. described the creation of an engineered probiotic strain, *Lactobacillus plantarum* WCSF I for monitoring the treatment of *S. aureus-associated* infections. The authors constructed an AgrQS system in *L. plantarum* capable of inhibiting *S. aureus* when detecting exogenous AIP nanomolar concentrations. According to my opinion, although the topic is interesting it is necessary to clarify some aspects and add some experiments to improve the paper as well as the understanding by the reader.

1. In the abstract the authors wrote that the engineering probiotics will be useful for monitoring intestinal pathogens. *S. aureus* is an opportunistic pathogen associated with a wide range of infections, therefore, why do not set up an engineered probiotic strain particularly versus Enterobacteriaceae (like *E. coli*, *Salmonella*) or *Listeria*, etc.? Please specify your choice.

✓ The treatment of many intestinal pathogens should be of great importance. Especially for Enterobacteriaceae (such as *Escherichia coli*, and *Salmonella*) or *Listeria*, However, specific recognition and detection signals for these bacteria have not been discovered yet and need to be further explored. Although *S. aureus* is not a typical intestinal pathogen, it can also cause intestinal inflammation in animals, which has potential research significance. We have revised and replied to this question in blue font in the fourth paragraph of the introduction.

2. Did the authors evaluate if the constructed could be subjected to the horizontal gene transfer? Please add some comments regarding this aspect.
 - ✓ We have responded to this question in blue font in the second paragraph of the discussion and seconded the literature to support it.
3. Did the authors evaluate a possible effect against other *Staphylococcus* species belonging to the microbiota? Please, test other *Staphylococcus* species to demonstrate the selectivity.
 - ✓ We have tested other *Staphylococcus* species, *Staphylococcus epidermidis* and *Staphylococcus scuri*, to verify the specificity of the constructed AgrQS gene circuitry. The result showed that engineered *L. plantarum* did not have a possible effect against these two *Staphylococcus* species, demonstrating that this AgrQS system cannot detect these two *Staphylococcus* species and it shows the selective specificity toward *S. aureus*. We have revised the manuscript in blue font in the third paragraph of the fourth part of the results and the specific results are presented in Figure 4d.
4. Did the authors evaluate a possible anti-biofilm activity of the supernatant produced by the *L. plantarum* modified strain? Please, demonstrate the quorum sensing inhibition.

In Part 5 of the results, we added controls for wild-type *L. plantarum* and *L. plantarum-Kill* (Figure. 5c and Figure. 5d) and added the anti-biofilm activity experiment of engineered *L. plantarum* supernatant on *S. aureus* biofilm (Figure. 5e) to verify the quorum sensing inhibition. We have revised it in blue font in the third paragraph of the fifth part of the results and described the specific operation of the experiment part in the material method.
5. Regarding the bacterial inhibition test (lines 295-305): the method is just a qualitative method. Is there a method to quantify the component associated with the inhibition (lysostaphin)?
 - ✓ To quantify the component of lysostaphin, the concentrations of lysostaphin secreted by engineered probiotics under different induction conditions were determined by the BCA protein concentration assay kit. We have revised it in blue font in the first paragraph of the fifth part of the results (Page 7, line 9 to line 11, and Page 8, line 14 to line 17).

6. In literature a similar approach has been used in *L. reuteri*. Please, add comments regarding the paper "Programming probiotic *L. reuteri* as a biosensor for *S. aureus* derived AIP 1 detection" by Lubkowicz et al. (2018).
 - ✓ We have added comments regarding the paper "Programming probiotic *L. reuteri* as a biosensor for *S. aureus* derived AIP-I detection" by Lubkowicz et al. (2018) in blue font in the second paragraph of the introduction.
7. Did the authors have any data regarding the safety of the modified strain on intestinal cell lines?
 - ✓ Thanks for your suggestions. Due to the limitations of conditions, we could not carry out the verification in animal experiments, but we found strong literature support that the engineered probiotics are safe for the application of animal intestinal tract. We have revised it in blue font in the first paragraph of the part of the discussion.
8. A more thorough treatment of the materials and methods would be appreciated for a better understanding by the reader. The section material and methods are very confusing.
 - ✓ We have revised and improved the writing of the Materials and Methods section.

Minor points:

1. Please, specify the number of CFU/ml corresponding to the OD concentration in the section material and methods.
 - ✓ We have revised and replied to this question in blue font in the first paragraph of the fifth part of the results (line 25 to line 26).
2. Please write in italics the bacterial genus and species in the text and the references.
 - ✓ We have corrected the genus and species of the species in the text and references in italics.
3. In the discussion, the authors should explain the possible clinical applications deriving from the use of the modified strain.
 - ✓ We have revised and replied to this question in blue font in the first paragraph of the discussion.
4. We have corrected the genus and species of the species in the text and references in italics.

- ✓ We have improved the quality of the images in the article and the supplementary documents.
- 5. Please add the number of independent experiments performed for the section bacterial inhibition test and *S. aureus* supernatant analysis.
- ✓ We have revised and replied to this question in blue font in the second paragraph of the fifth part of the results. We recorded the diameter of inhibiting *S. aureus* production after several tests to verify the antibacterial effect of engineered *L. plantarum* (Figure 5d).

September 23, 2023

Prof. Qian Wang
Shandong University - Qingdao Campus
Life Science School
27, shanda south road
571, north building, Life Science School
Jinan, Shandong 250100
China

Re: Spectrum01829-23R1 (Engineered probiotic *Lactobacillus plantarum* WCSF I for monitoring and treatment of *Staphylococcus aureus* infection)

Dear Prof. Qian Wang:

Your manuscript has been accepted, and I am forwarding it to the ASM Journals Department for publication. You will be notified when your proofs are ready to be viewed.

Sincerely,

Harold Marcotte
Editor, Microbiology Spectrum
